# Long Range Graph Benchmark

**Vijay Prakash Dwivedi**[*]
Nanyang Technological University, Singapore

**Ladislav Rampášek**
Mila, Université de Montréal

**Mikhail Galkin**
Mila, McGill University

**Ali Parviz**
NJIT, Mila, Université de Montréal

**Guy Wolf**
Mila, Université de Montréal

**Anh Tuan Luu**
Nanyang Technological University, Singapore

**Dominique Beaini**
Valence Discovery, Mila, Université de Montréal

## Abstract

Graph Neural Networks (GNNs) that are based on the message passing (MP) paradigm generally exchange information between 1-hop neighbors to build node representations at each layer. In principle, such networks are not able to capture long-range interactions (LRI) that may be desired or necessary for learning a given task on graphs. Recently, there has been an increasing interest in development of Transformer-based methods for graphs that can consider full node connectivity beyond the original sparse structure, thus enabling the modeling of LRI. However, MP-GNNs that simply rely on 1-hop message passing often fare better in several existing graph benchmarks when combined with positional feature representations, among other innovations, hence limiting the perceived utility and ranking of Transformer-like architectures. Here, we present the Long Range Graph Benchmark (LRGB)[1] with 5 graph learning datasets: `PascalVOC-SP`, `COCO-SP`, `PCQM-Contact`, `Peptides-func` and `Peptides-struct` that arguably require LRI reasoning to achieve strong performance in a given task. We benchmark both baseline GNNs and Graph Transformer networks to verify that the models which capture long-range dependencies perform significantly better on these tasks. Therefore, these datasets are suitable for benchmarking and exploration of MP-GNNs and Graph Transformer architectures that are intended to capture LRI.

## 1 Introduction

Considering a graph as a collection of nodes where arbitrary relations between nodes are represented as edges, there are numerous real-world instances with data structures where complex and irregular interactions among objects can be represented as edges where the objects themselves are denoted as nodes. This has led to a rapid rise of interest in the development of graph neural networks (GNNs) [24, 42, 9, 61] for deep learning on geometric and graph domains.

The popularly used class of GNNs is based on the message passing paradigm [22] where a node's feature representation, at each layer, is updated using a trainable function that receives and combines feature information from all its neighboring nodes. A network that has $L$ GNN layers, stacked sequentially, can iteratively aggregate feature information from up to $L$ hops for the update of a node's representation. If a long-range information to a node from its $L$-hop neighbor is needed for a task (say, for a large $L$), the same number of GNN layers is ideally required. However, with the

---

[*]To whom correspondence should be addressed: `vijaypra001@e.ntu.edu.sg`

[1]Open-sourced at `https://github.com/vijaydwivedi75/lrgb` and deposited at Zenodo [16].

36th Conference on Neural Information Processing Systems (NeurIPS 2022) Track on Datasets and Benchmarks.

Table 1: Overview of the datasets in the proposed LRGB. Note: 'Pixels+Coord' denotes the feature vector consisting of 12-dim statistics of each superpixel (for each RGB color: the average, standard deviation, maximum, and minimum of pixel intensities in the superpixel) and 2-dim coordinates of the center of mass of its X,Y pixel locations. 'Edge Weight' corresponds to the weight assigned between two superpixel nodes w.r.t. the construction method. The 'Atom Encoder' and 'Bond Encoder' are OGB molecular feature encoders [27, 26]. All tasks are inductive tasks.

| Dataset | Domain | Task | Node Feat. (dim) | Edge Feat. (dim) | Perf. Metric |
|---|---|---|---|---|---|
| PascalVOC-SP COCO-SP | Computer Vision | Node Classif. | Pixel + Coord (14) | Edge Weight (1 or 2) | macro F1 |
| PCQM-Contact | Quantum Chemistry | Link Prediction | Atom Encoder (9) | Bond Encoder (3) | Hits@K, MRR |
| Peptides-func Peptides-struct | Chemistry | Graph Classif. Graph Regression | Atom Encoder (9) | Bond Encoder (3) | AP MAE |

Table 2: Statistics of the five proposed LRGB datasets.

| Dataset | Total Graphs | Total Nodes | Avg Nodes | Mean Deg. | Total Edges | Avg Edges | Avg Short.Path. | Avg Diameter |
|---|---|---|---|---|---|---|---|---|
| PascalVOC-SP | 11,355 | 5,443,545 | 479.40 | 5.65 | 30,777,444 | 2,710.48 | 10.74±0.51 | 27.62±2.13 |
| COCO-SP | 123,286 | 58,793,216 | 476.88 | 5.65 | 332,091,902 | 2,693.67 | 10.66±0.55 | 27.39±2.14 |
| PCQM-Contact | 529,434 | 15,955,687 | 30.14 | 2.03 | 32,341,644 | 61.09 | 4.63±0.63 | 9.86±1.79 |
| Peptides-func | 15,535 | 2,344,859 | 150.94 | 2.04 | 4,773,974 | 307.30 | 20.89±9.79 | 56.99±28.72 |
| Peptides-struct | 15,535 | 2,344,859 | 150.94 | 2.04 | 4,773,974 | 307.30 | 20.89±9.79 | 56.99±28.72 |

increasing $L$ the size of the $L$-hop neighborhood grows exponentially and so does the amount of information that needs to be encoded into one vector by the network. This leads to 'information oversquashing' as the message aggregation step continues to be iteratively applied at each layer in order to propagate the information [2]. Consequently, such GNNs fail at capturing long-range dependencies as a significant amount of distant information may get lost due to the squashing.

In order to factor in distant information when message passing GNNs are used, Alon et al. [2] used a fully connected graph at the final layer as an intuitive remedy. The primary rationale behind this approach is to enable each node in a graph to connect to every other node at some stage in the network to pass the information that otherwise would get squashed, thus breaking the bottleneck. Consequently, several recent works propose Graph Transformers that leverage full-connections among all nodes in the graph to capture long-range dependencies [34, 65, 44].

However, it is often the case that these models are evaluated on datasets where the corresponding tasks primarily rely on local structural information rather than the distant information propagation between nodes. This observation is prevalent for many existing datasets such as ZINC [14], ogbg-molpcba, or ogbg-molhiv [27] that are among the most frequently used benchmarks. The Spectral Attention Network (SAN) [34] has shown insignificant contribution of full attention in these benchmarks. In fact, leaderboards of these benchmarks are topped by local MP-GNN based models [3, 68, 37], albeit these GNNs are non trivial extensions and are augmented with higher-order structural information, among other model improvements. At the same time, these molecular benchmarks largely consist of graphs of small sizes, *i.e.*, the number of nodes in a graph. Nonetheless, on the contrary, graphs with large number of nodes may not necessarily imply that they require models with long-range dependencies for the learning task.

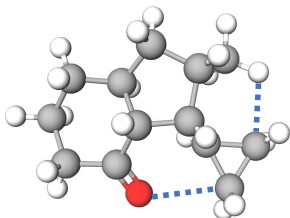

Figure 1: Molecule with LRIs (dotted lines showing 3D atomic contact) that are not trivially captured by the graph structure.

**Contribution.** In this work, we focus on these shortcomings of existing popular graph learning benchmark datasets and propose characterizing factors in a dataset that can be studied for the exploration of new GNN and Graph Transformer architectures that possess long-range interaction (LRI) capabilities. Note that our characterization henceforth is not directed at proposing 'provable LRI' benchmarks, which would often lead to toy datasets (that are useful for quick prototyping of ideas) such as the shortest path prediction task [55] or the color connectivity dataset [50] which rely on LRI. Instead, our aim is to propose real-world datasets that require LRI, and the factors we consider for a LRGB dataset characterization could be understood as implications which suggest that

the learning task(s) in the graphs would depend on long range signal propagation. Consequently, we introduce 5 benchmarking datasets – `PascalVOC-SP`, `COCO-SP`, `PCQM-Contact`, `Peptides-func` and `Peptides-struct` from the domains of Computer Vision and Chemistry which we incorporate in LRGB, see Tables 1-2 for an overview and Figure 1 for a sample illustration. The learning tasks that we propose in these datasets depend on some degree of long-range signal handling given the nature of task, contribution of global graph structure to the task, and the sizes of graphs in these datasets. Fittingly, in our baseline experiments, these datasets show that the fully-connected models which enable LRI propagation perform considerably better than local message passing based GNNs.

**Existing attempts towards LRI benchmarks.** Thanks to the understanding of the limitations of message passing based GNNs [22] with respect to the 1-Weisfeiler Leman (WL) isomorphism test [60, 62, 46] and the information oversquashing [2], there has been several developments of GNN and Graph Transformer architectures which have strictly greater representation power than 1-WL [7, 3, 46, 45, 35, 34, 65]. By design, fully connected Graph Transformers [65, 34, 44] are able to model long-range dependencies in the graphs and alleviate information bottleneck to some extent [53]. Similarly, some recent models have been designed to perform non-local feature integration in particular aspect of non-homophilic graphs [49, 40]. However, most of such architectures are evaluated on benchmarks where it is not clear whether long-range interactions are required for the corresponding learning tasks. For natural language processing (NLP), the Long Range Arena [57] benchmark has been instrumental in studying the capacity and efficiency of architectures against longer sequences. Notably, a recent work has introduced non-homophilic graph datasets [36] that is an orthogonal attempt to contribute towards graph learning testbeds beyond the widely used benchmarks which favor non-local GNN methods.

Nevertheless, we believe there is a consensus in the community towards the development of specific benchmarks that can assist LRI enabled-GNNs, including full-graph operable Graph Transformers. This can be observed in existing independent attempts at proposing new graph benchmarks to evaluate LRI. In Stachenfeld et al. [55], a new Graph MNIST benchmark with an increased average graph diameter compared to MNIST superpixels [12, 14] was used to evaluate the ability of Spectral Graph Network in incorporating long range signals. Similarly, a synthetic color connectivity task that, by construction, requires LRI to differentiate between its classes was used in Rampášek & Wolf [50] to demonstrate a hierarchical graph network's ability in modeling such signals. Another synthetic benchmark was used in Alon et al. [2] to probe the oversquashing phenomenon and implement intuitive tricks such as a fully connected graph layer to alleviate the bottleneck. Finally, a Chains dataset [23] was created for testing long-range dependency that was adopted to develop enhanced models such as higher order Transformers [31], among others [64, 39]. Apart from the aforementioned synthetic and semi-real tasks, MalNet [20], a real-world dataset of large function call graphs (avg. 15k nodes) was recently proposed. MalNet could be a potential LRI task given its graph sizes, which we discuss in Section 2 as a characterizing aspect of LRI benchmarks, along with other factors.

## 2 Characterizing Long-Range Interactions

We now proceed to study the key characteristics that can help to determine how a graph dataset could be appropriate to guage whether a GNN can or cannot model LRIs. Note that our discussion here is based on datasets with inductive tasks, which contain many graphs, rather than a single (large) graph.

**Graph Size.** The number of nodes in a graph is critical to determine if any visible effect of information oversquashing would occur if a local message passing based GNN (MP-GNN) is used to learn on this graph. If $r$ is a hypothetical estimate of the learning problem's radius in the graph or the problem's range of interaction [2], and $L \geq r$ denotes the number of layers that are stacked in a GNN to learn the task, the number of nodes in a node's receptive field grows exponentially, *i.e.*, $O(\exp(L))$ [2, 10]. However, if the graph size is small, such as `ogbg-mol*` [27] or ZINC [14] datasets with average graph size in the range of 23-26 nodes, then $r$ would effectively be small as well. As a consequence, the effect of squashing of the information from the node's receptive field will be diminishing, and any local MP-GNN would succeed to learn the task to a great extent without being influenced by the information bottleneck. Therefore, a direct conclusion of this condition is that for a LRI benchmark, the graph sizes should be sufficiently large in order to separate the local MP-GNNs' performance from those models which model LRIs. However, this condition of graph size alone may not be enough to determine a LRGB dataset as the problem radius $r$ may be small for some tasks even if the graph size is large, which brings us to the following factors.

**Nature of Task.** The nature of task can be understood to be directly related to the problem's range of interaction, $r$. In broad sense, the task can be *either* short-range, *i.e.*, requiring information exchange among nodes in local or near-local neighborhood, *or* long-range, where interactions are required far away from the near-local neighborhood. For instance, the task in the ZINC molecular dataset [29, 14] is associated with counting local structures and it has been revealed that a substructure-counting based model [7] would optimally require counts of 7-length substructures for the best performance. Any increment above this length does not further show a gain in the ZINC task. It may therefore be interpreted that such a benchmarking task does not require long-range signal propagation. Additionally, note that the graph sizes in ZINC are small (9-37 nodes) which functionally makes it a non-LRI benchmark if we also factor in the nature of ZINC's task.

However, even if the graph size of a dataset is considerably large, it may not warrant that models with long-range signal propagation are best suited unless the nature of the task determines so. A recent example to this is MalNet-Tiny dataset [20] consisting of graphs up to 5,000 nodes, where there is a scarce improvement of performance of fully-connected GNN modules [51]. Finally, there exist tasks in graphs which are prone to bottleneck if local MP-GNNs are used while this bottleneck is substantially reduced if LRI enabled non-local MP-GNNs are used, see Table 3 in Shi et al. [53].

**Contribution of global graph structure to task.** Since MP-GNNs rely on information aggregated from a local neighborhood to update a node's features, it is subject to miss global structural information, such as global positional encoding (PE) [51]. Additionally, MP-GNNs are also susceptible to lose out critical node signals coming from distant nodes if the graph size is large enough [2]. Such signals are conveniently propagated in a fully-connected Transformer-like networks modeling LRI. The contribution of global structure to a task thus becomes a distinctive property desired in a LRI benchmark. MP-GNNs are often augmented with positional encodings (PE) carrying global structural information to assist tasks requiring some degree of LRI. A dataset where the learning task benefits from global PE can hence be a potential LRI benchmark. Similarly, if the learning task in a dataset is dependent on some form of distance information, or is directly a function of distance, coupled with graph feature information, the dataset can be a strong candidate for LRGB since the distance information would require global structural information. Examples of this can be molecular datasets where the learning task is related to prediction of 2D or 3D distance and structure properties.

## 3 Proposed LRGB Datasets

### 3.1 `PascalVOC-SP`

`PascalVOC-SP` is a node classification dataset, based on the Pascal VOC 2011 image dataset [18], where each node corresponds to a region of the image belonging to a particular class. The original dataset is available on a Custom License (respecting Flickr terms of use) [48]. Similar to the recent superpixels (SP) datasets such as MNIST and CIFAR10 [14], we extract superpixels nodes in `PascalVOC-SP` by using the SLIC algorithm [1] and construct a `rag-boundary` graph that interconnects these nodes. Unlike MNIST and CIFAR10 superpixels which have up to 75 and 150 nodes respectively, we extract a maximum of 500 superpixel nodes for SLIC compactness value of 30[2] in `PascalVOC-SP` in order to satisfy the 'graph size' characteristic to make it a LRGB benchmark. Effectively, it results in the `PascalVOC-SP` dataset to have an average shortest path length of 10.74$\pm$0.51 and average diameter of 27.62$\pm$2.13 (see Table 2) which is significantly larger than that of MNIST with 3.03$\pm$0.17, 6.03$\pm$0.47 and CIFAR10 with 3.97$\pm$0.08, 8.46$\pm$0.50 average shortest path and diameters respectively. We argue that these properties, along with the task of predicting the node label of the superpixel region, which is analogous to the semantic segmentation task in Computer Vision, makes `PascalVOC-SP` a suitable LRGB dataset fulfilling major characteristics discussed in Section 2. We also prepare other variants of `PascalVOC-SP` with different values of SLIC compactness and graph construction options, which are included in Appendix A.

**Statistics.** There are 11,355 graphs with a total of 5.4 million nodes in `PascalVOC-SP` where each graph corresponds to an image in Pascal VOC 2011. The graphs prepared after the superpixels extraction have on average 479.40 nodes with complete statistics reported in Table 2.

---

[2]The compactness parameter balances spatial and color information when extracting superpixels in SLIC [1].

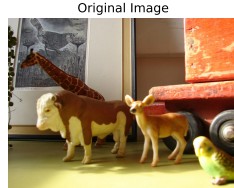 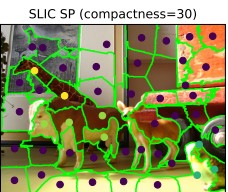 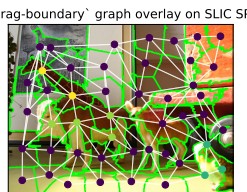 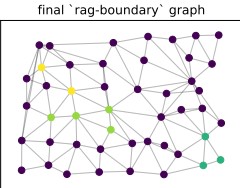

| Original Image | SLIC SP (compactness=30) | `rag-boundary` graph overlay on SLIC SP | final `rag-boundary` graph |

Figure 2: Visualization of a sample image, its SLIC SP regions and `rag-boundary` graph from `COCO-SP` dataset. In this figure, the extracted SP are $< 50$ for better visualization. For the actual graphs with a maximum of $500$ SP and nodes in `PascalVOC-SP` and `COCO-SP`, refer to Appendix B.

**Task.** The task in `PascalVOC-SP` is node classification which predicts a semantic segmentation label for each superpixel node out of 21 classes. We label each superpixel node with the same class label of the original pixel ground truth which is on the mean coordinates of the superpixel region.[3]

**Splitting.** In the original Pascal VOC 2011 dataset, there are only training and validation splits that we can use. For `PascalVOC-SP`, we maintain the train set as it is, and split the original validation set into new validation and test sets. For this splitting, we divide the original validation set in 50:50 ratio using a stratified split proportionate to the original distribution of the data with respect to a meta label that depends on the node classes. This meta label is a ground truth class value obtained by a majority voting of non-background ground truth node labels. The splitting decision with this meta-label is taken to preserve a similar distribution of node labels in both the new validation and the test set. Thus, we have 8,498, 1,428 and 1,429 graphs in the final training, validation and test sets, respectively.

**Construction.** After the superpixels extraction, we prepare the `rag-boundary` graph as illustrated step-wise in Figure 2. Two superpixels nodes are connected with an edge if the node regions share a common boundary. We use the `rag_boundary` functionality from `skimage` [6] to extract the region boundaries. By construction, the dataset in this `rag-boundary` graph format has nodes with varying number of neighbors. This construction format also makes the graph more sparse with an average node degree of 5.6 for graphs averaging 479.40 node sizes. The initial feature of each superpixel node is 14 dimensional, 12-dim RGB feature value (mean, std, max, min) and 2-dim coordinates of the center of mass of pixel locations, and that of an edge between two nodes is a 2 dimensional vector where the first value is 'weight' denoting the average of the Sobel filter [17] pixel values along the boundary between the 2 adjacent regions, and the second value is 'count' denoting the count of all pixels along this boundary.

**Performance Metric.** The performance metric is the macro weighted F1 score for the predicted node label and the ground truth node label.

### 3.2 `COCO-SP`

Similar to `PascalVOC-SP`, `COCO-SP` is a node classification dataset based on the MS COCO image dataset [38] where each superpixel node denotes an image region belonging to a particular class. The original MS COCO image dataset is available under CC BY 4.0 License. We follow the same steps as in Section 3.1 for the preparation of superpixels and the graphs in `COCO-SP` for the `rag-boundary` graph format. Additional optional variants of the `COCO-SP` datasets are included in Appendix A.

**Statistics.** There are 123,286 graphs with a total of 58.7 million nodes in `COCO-SP` where each graph corresponds to an image in MS COCO dataset [38]. The graphs prepared after the superpixels extraction have on average 476.88 nodes with complete statistics reported in Table 2.

**Task.** The learning task in `COCO-SP` is node classification to predict a semantic segmentation label for each superpixel node out of 81 classes. We label each superpixel node with the same class label of the original pixel ground truth which is on the mean coordinates of the superpixel region.

**Splitting.** In the MS COCO image dataset there are only train and validation sets available that we can use. In `COCO-SP`, we maintain the original validation set as the new test set, while we sample 5,000 images from the original training set to generate the new validation set. Finally, there are 113,286 graphs, 5,000 graphs and 5,000 graphs in the resultant training, validation and test set, respectively.

---

[3]The labels are based on the annotations provided in Semantic Boundary Dataset (SBD) version of Pascal VOC 2011: `https://github.com/shelhamer/fcn.berkeleyvision.org/tree/master/data/pascal`

**Performance Metric.** Similar to `PascalVOC-SP`, the performance metric is the macro weighted F1 score for the predicted node label and the groundtruth node label.

### 3.3  `PCQM-Contact`

Molecular property prediction is one of the most popular tasks for benchmarking GNNs. The usual task is to predict a biochemical property [27, 14]. Molecular datasets are very interesting for the study of Graph Transformers since their properties do not only depend on local graph structure defined by covalent bonds, but *inter alia* also on long-range interactions that define the 3D folding of the molecules, their surface area, or the energy of their electronic orbitals [66, 41, 56].

However, existing benchmarks do not necessarily depend on long-range interactions and are noisy to properly evaluate Graph Transformers. For instance, the task in ZINC dataset from Dwivedi et al. [14] depends on a linear combination of local structures [30]. Thus, it is unclear whether there is any benefit in using a fully-connected Transformer instead of a standard message passing network [51]. OGB [27] offers a variety of datasets from biological assays, but they are often small, noisy, and it is unclear whether they would benefit from the long-range interactions of a Transformer. In `PCQM-Contact`, we design a task that explicitly requires LRI since it needs to understand the interaction between distant atoms.

**Statistics.** There are 529,434 graphs with a total of 15 million nodes in `PCQM-Contact` where each graph corresponds to a molecular graph with explicit hydrogens, and more details in Table 2. All graphs were taken from the PCQM4M training set with available 3D structure [26] and filtered to only keep those with at least one contact.

**Task.** The task is to predict pairs of distant nodes (more than 5 hops away from each other in a molecule graph) that will be contacting with each other in the 3D space, i.e., the 3D distance between atoms will be smaller than 3.5Å. The threshold of 3.5Å is chosen to account for hydrogen bonds, one of the most common non-covalent interaction with a typical distance of 2.7 to 3.3 Å [43]. The 5-hop distance is chosen to avoid *trivial* predictions between atoms that are close in the molecular graph and force the network to learn properties related to the 3D structure. Note that, contrarily to most benchmarks that represent hydrogens implicitly with node features, `PCQM-Contact` makes the hydrogens atoms explicit by adding a node.

Contact map prediction is therefore framed as inductive link prediction. That is, training molecules only have true positive links without hard negative labels whereas at validation and test time we predict contact links over new, unseen molecules. The molecules are treated as relational graphs with learnable edge types (standard BondEncoder in OGB [26] e.g., single bond, double bond, triple bond), but the predictable contact link does not have an explicit edge type, so a link prediction decoder is a function $f(h, t)$ of probed head and tail nodes.

**Splitting.** We randomly split the dataset into 90% (476,490 molecules) training split, 5% (26,472 molecules) validation split, and 5% (26,472 molecules) testing split.

**Performance Metric.** In the absence of true negatives, we resort to the ranking metrics common in the knowledge graph link prediction literature [4]. Given a query $(h, ?)$, we compute a scalar score for each other node in a graph as a tail $(h, t_i)$, and look for a rank of a true positive link. The true link has rank 1 if its score is the highest among all other links. We use standard ranking metrics Hits@1, Hits@3, and Mean Reciprocal Rank (MRR, *aka* Inverse Harmonic Mean Rank [25]) in the *filtered* setting [4]. That is, if there exist several true links sharing the same head (or tail), i.e., $(h, t_1), (h, t_2), \ldots, (h, t_k)$, evaluating each link separately, we filter out (mask) scores of other true tails setting their scores to $-\infty$ such that they do not interfere with the ranking procedure.

### 3.4  Peptides molecular graphs

Peptides are short chains of amino acids that are abundant in nature as they serve many important biological functions [54], but they are much shorter than proteins [47]. Since each amino acid is composed of many heavy atoms, the molecular graph of a peptide is much larger than that of a small drug-like molecule. Peptides have about 6 times large diameter and 5 times more atoms than the `PCQM-Contact` dataset, but similar avg. degree of ~2. See Figure 3 for an illustration. This makes them ideal for testing long-range dependencies in GNNs while still being able to fit an entire mini-batch on a single GPU.

Here we propose `Peptides-func` and `Peptides-struct` datasets, derived from 15,535 peptides retrieved from SATPdb [54]. Both datasets use the same set of graphs but differ in their prediction tasks.

**Construction.** The graphs are derived such that the nodes correspond to the heavy (non-hydrogen) atoms of the peptides while the edges represent the bonds between them. We reuse the OGB molecular featurization [27] that computes rich node and edge features from molecular SMILES.

In both `Peptides-func` and `Peptides-struct`, recognizing local structures is very important for the model to even identify the original amino acids. Further, we do not include any 2D or 3D peptide structure information. The graphs correspond to 1D amino acids chains, which means it is important for the model to identify the location of an amino acid in the graph. Finally, with the peptides chains having different lengths and a strong variability in their graph diameters, any used graph positional or structural encoding needs to generalize well across various sizes and be computationally efficient.

**Statistics.** Both `Peptides-func` and `Peptides-struct` consist of 15,535 graphs with a total of 2.3 million nodes, Table 2. All peptides were obtained from the SATPdb [54] database (an aggregate of multiple public-domain sources) that includes the sequence, molecular graph, function, and 3D structure of the peptides.

Previously introduced ENZYMES and PROTEINS datasets [5], that use the 3D structure of the folded proteins to build a graph of amino acids, are notably different from those that we propose here. In addition to more complex prediction tasks, our datasets are also larger in multiple ways. First, we derive 15,535 graphs, compared to theirs 600 and 1,113, respectively. Second, we use heavy atoms as nodes and not the amino acids, resulting in larger graphs: on average 150.94 nodes per graph, compared to theirs 32.63 and 39.06, respectively. In terms of graph diameter, our graphs average 56.99 compared to theirs 10.92 and 11.62, respectively. Thus, the proposed `Peptides-func` and `Peptides-struct` are better suited to benchmarking of graph Transformers or other expressive GNNs, as they contain larger graphs, more data points, and challenging tasks.

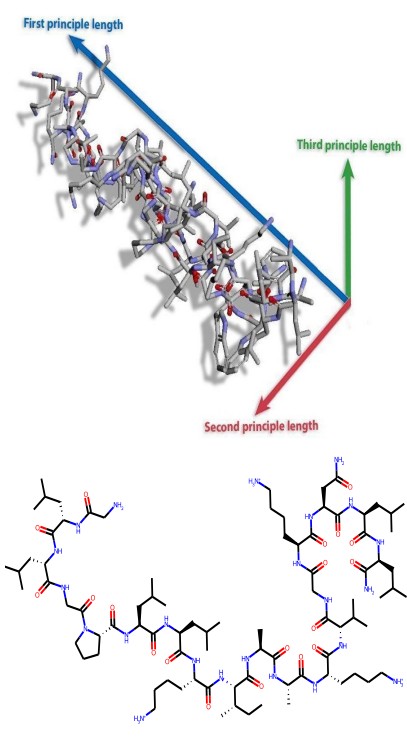

Figure 3: **Top:** 3D Visualization of "GLLGPLLKIAAKVGKNLL" peptide. **Bottom:** The molecular graph for the same peptide.

### 3.4.1 `Peptides-func`

**Task.** `Peptides-func` is a multi-label graph classification dataset. There is a total of 10 classes based on the peptide function, e.g., *Antibacterial, Antiviral, cell-cell communication*, and others. We treat it as a multi-label classification as a peptide can belong to several classes simultaneously; on average to 1.65 of the 10 classes. The labels are imbalanced, only 16.5% of the data is in the positive class, with the richest class having 62.7% positives and the poorest 1.9%. The correlation between individual classes is shown in Figure A.1.

**Splitting.** For the purpose of split computation, the data is first aggregated into meta-classes by considering the concatenation of all 10 original labels of a data point as its meta-class. Meta-classes with less than 10 occurrences are pooled into one meta-class. Then we apply stratified splitting to generate balanced train–valid–test dataset splits; we use the ratio of 70%–15%–15%, respectively.

**Performance Metric.** We choose the unweighted mean Average Precision (AP). This metric measures the area under the *precision-recall curve*, and is also used for `ogbg-molpcba` in OGB, a dataset with similar imbalanced multi-label classification.

Table 3: Baseline experiments for `PascalVOC-SP` and `COCO-SP` with `rag-boundary` graph on SLIC compactness 30 for node classification task (Extended results for all graph formats in Table A.2). Performance metric is macro F1 on the respective splits (Higher is better). All experiments are run 4 times with 4 different seeds. The MP-GNN models are 8 layers deep, while the transformer-based models have 4 layers in order to maintain comparable hidden representation size at the fixed parameter budget of 500k. *The SAN model under-fitted the `COCO-SP` dataset since it required more budget than the 60 hours allowed on A100 GPUs. **Bold**: Best score.

| Model | # Params | PascalVOC-SP | | # Params | COCO-SP | |
|---|---|---|---|---|---|---|
| | | Train F1 | Test F1 ↑ | | Train F1 | Test F1 ↑ |
| GCN | 496k | 0.1450±0.0125 | 0.1268±0.0060 | 509k | 0.0948±0.0014 | 0.0841±0.0010 |
| GCNII | 492k | 0.2272±0.0245 | 0.1698±0.0080 | 505k | 0.2020±0.0127 | 0.1404±0.0011 |
| GINE | 505k | 0.2088±0.0268 | 0.1265±0.0076 | 515k | 0.2100±0.0041 | 0.1339±0.0044 |
| GatedGCN | 502k | 0.3552±0.0451 | 0.2873±0.0219 | 509k | 0.3167±0.0059 | **0.2641±0.0045** |
| GatedGCN+LapPE | 502k | 0.3512±0.0167 | 0.2860±0.0085 | 509k | 0.3102±0.0112 | 0.2574±0.0034 |
| Transformer+LapPE | 501k | 0.7170±0.0048 | 0.2694±0.0098 | 508k | 0.3912±0.0098 | 0.2618±0.0031 |
| SAN+LapPE | 531k | 0.5723±0.0427 | **0.3230±0.0039** | 536k | 0.2830±0.0246* | 0.2592±0.0158* |
| SAN+RWSE | 468k | 0.5819±0.0331 | 0.3216±0.0027 | 474k | 0.2657±0.0224* | 0.2434±0.0156* |

### 3.4.2 `Peptides-struct`

**Task.** `Peptides-struct` is a multi-label graph regression dataset based on the 3D structure of the peptides. It consists of the same graphs as `Peptides-func`, but with different task. Here we aim to predict aggregated 3D properties of the peptides at the graph level. The properties (normalized to zero mean and unit standard deviation) include: *Inertia_mass*, *Inertia_valence*, *Length*, *sphericity*, and *plane_best_fit*. How we derive these properties is described in Appendix A.2 and their correlations are shown in Figure A.1. These new tasks are expected to directly benefit from the full-connectivity of a Transformer, because they require implicit understanding of complex 3D interactions. To adapt these tasks to the general graph learning setting, we avoided the prediction of pairwise node distances, which would require specialized methods from the conformer generation literature [21, 52, 63].

**Splitting.** The data splits are identical to `Peptides-func`. Since structure is related to functionality, this also ensures that the structures in the testing set represent well the training and validation sets.

**Performance Metric.** For the `Peptides-struct` dataset, we use the Mean Absolute Error (MAE), typically selected for molecular property regression datasets. We further track the Coefficient of Determination ($R^2$) on each task, and compute its unweighted mean across tasks.

## 4 Experiments and Discussion

### 4.1 Baseline experiments

We conduct baseline experiments on our proposed LRGB datasets by training and evaluating two GNN classes: (i) local MP-GNNs, and (ii) fully connected Graph Transformers. We adopt fair and rigorous experimental settings for all our experiments in order to present reliable comparison between the two GNN classes. The former models do not directly include any mechanism to model LRI, while the latter are by design fully connected and can propagate long-range signals, which are required for the proposed benchmarks. For the baselines, we select GCN [33], GCNII [11], GINE [62, 28] and GatedGCN [8] models from the local MP-GNN class, and fully connected Transformer [58] with Laplacian PE (LapPE) [14, 13] and SAN [34] models from the Transformer class. In order to facilitate fair comparison and reliable discussion of the observed trends, we choose hyperparameters of the aforementioned baselines while keeping to a budget of 500k learnable parameters. Detailed experimental setup and hyperparameters are provided in Appendix C.

### 4.2 Results and Analysis

The baseline results for `PascalVOC-SP` and `COCO-SP` benchmarks are reported in Table 3, for `Peptides-func` and `Peptides-struct` in Table 4, and for `PCQM-Contact` in Table 5. We also

Table 4: Baselines for `Peptides-func` (graph classification) and `Peptides-struct` (graph regression). Performance metric is Average Precision (AP) for classification and MAE for regression (see Table A.3 for extended results with $R^2$ metric). Each experiment was run with 4 different seeds. All MP-GNN models have 5 layers, while the Transformer-based models have 4 layers. **Bold**: Best score.

| Model | # Params. | Peptides-func | | Peptides-struct | |
| --- | --- | --- | --- | --- | --- |
| | | **Train AP** | **Test AP ↑** | **Train MAE** | **Test MAE ↓** |
| GCN | 508k | 0.8840±0.0131 | 0.5930±0.0023 | 0.2939±0.0055 | 0.3496±0.0013 |
| GCNII | 505k | 0.7271±0.0278 | 0.5543±0.0078 | 0.2957±0.0025 | 0.3471±0.0010 |
| GINE | 476k | 0.7682±0.0154 | 0.5498±0.0079 | 0.3116±0.0047 | 0.3547±0.0045 |
| GatedGCN | 509k | 0.8695±0.0402 | 0.5864±0.0077 | 0.2761±0.0032 | 0.3420±0.0013 |
| GatedGCN+RWSE | 506k | 0.9131±0.0321 | 0.6069±0.0035 | 0.2578±0.0116 | 0.3357±0.0006 |
| Transformer+LapPE | 488k | 0.8438±0.0263 | 0.6326±0.0126 | 0.2403±0.0066 | **0.2529±0.0016** |
| SAN+LapPE | 493k | 0.8217±0.0280 | 0.6384±0.0121 | 0.2822±0.0108 | 0.2683±0.0043 |
| SAN+RWSE | 500k | 0.8612±0.0219 | **0.6439±0.0075** | 0.2680±0.0038 | 0.2545±0.0012 |

Table 5: Baseline performance on `PCQM-Contact` (link prediction). Each experiment was repeated with 4 different random seeds. The evaluated models have 5 (MP-GNN models) or 4 (Transformer-based models) layers with approximately 500k learnable parameters. **Bold**: Best score.

| Model | # Params. | Test Hits@1 ↑ | Test Hits@3 ↑ | Test Hits@10 ↑ | Test MRR ↑ |
| --- | --- | --- | --- | --- | --- |
| GCN | 504k | 0.1321±0.0007 | 0.3791±0.0004 | 0.8256±0.0006 | 0.3234±0.0006 |
| GCNII | 501k | 0.1325±0.0009 | 0.3607±0.0003 | 0.8116±0.0009 | 0.3161±0.0004 |
| GINE | 517k | 0.1337±0.0013 | 0.3642±0.0043 | 0.8147±0.0062 | 0.3180±0.0027 |
| GatedGCN | 527k | 0.1279±0.0018 | 0.3783±0.0004 | 0.8433±0.0011 | 0.3218±0.0011 |
| GatedGCN+RWSE | 524k | 0.1288±0.0013 | 0.3808±0.0006 | 0.8517±0.0005 | 0.3242±0.0008 |
| Transformer+LapPE | 502k | 0.1221±0.0011 | 0.3679±0.0033 | 0.8517±0.0039 | 0.3174±0.0020 |
| SAN+LapPE | 499k | **0.1355±0.0017** | 0.4004±0.0021 | 0.8478±0.0044 | **0.3350±0.0003** |
| SAN+RWSE | 509k | 0.1312±0.0016 | **0.4030±0.0008** | **0.8550±0.0024** | 0.3341±0.0006 |

provide additional baseline results for all MP-GNNs with fewer layers ($L = 2$) in Appendix D. Our aim is to address the following main questions through the analysis of these results:

(i) Is a local feature aggregation, modeled using MP-GNNs with fewer layers, enough for the proposed tasks in LRGB?

(ii) Do we observe a visible separation in learning and generalization of models with enhanced capability to capture LRIs when compared against local MP-GNNs on the proposed benchmark?

(iii) Does the use of positional encodings, that contribute critical structural information, improve MP-GNN performance on the proposed datasets?

(iv) What are the challenges and future discoveries that can be facilitated by the new benchmarks?

**Simple instances of local MP-GNNs perform poorly on the proposed LRGB datasets.** As shown by the results in Tables 3, 4 and 5, GCN and GINE, which depend on local feature aggregation from node neighborhoods using simple aggregation functions, perform poorly on all datasets except `Peptides-func`. This is consistent with the empirical findings in [2] where GCN and GIN suffer from over-squashing to a greater extent than GAT, an attention based MP-GNN [59].

**Shallow MP-GNNs that gather information from only close neighbors underfit.** The comparison of shallow MP-GNN baselines ($L = 2$, see Appendix D) with deeper ones ($L = 5, 8$, see Tables 3-5) shows, that models with information aggregation limited to only a few hops significantly underfit and provide poor generalization on test set, as compared to MP-GNNs with the increased receptive field. This points towards the proposed benchmarks being *different* from several classical benchmarks such as Cora or Citeseer, *inter alia* [11], where shallow GCNs [33] fared better than deeper GCNs. They are thus more suitable for evaluating GNNs with deeper architectures, increased receptive fields, as well as long-range modeling, for which we provide a further study in the following analysis.

**Transformers operating on fully-connected graph show the best performance.** It can be observed that the Transformer model and the SAN, which is an improved Transformer, rank among the best

performing baselines in Tables 3, 4 and 5, except for GatedGCN vs. Transformer in Table 3. The gap in performance seems the most distinct for `Peptides-func` and `Peptides-struct` among all datasets. This can be attributed to the long-range design of the task, the contribution of non-local information, and the graph size statistics of these datasets, as discussed in Section 3.4. On `PascalVOC-SP` and `PCQM-Contact`, SAN performs comparatively better than vanilla Transformer+LapPE, suggesting that full connections in graphs should be used in non-trivial manner for LRI enabled models to do well. Finally, on `COCO-SP` the observed difference between Transformer and GatedGCN is minor, while SAN did not manage to converge within a 60h computational limit. This exposes the scalability drawbacks of current Transformer-based models, that would likely benefit from increased parameter and computational budget, see Table C.2. In Appendix E, we show additional investigation, with visualizations, on how Transformer exhibits attention patterns beyond local neighborhoods in general.

**Discussion on LRI characterizing factors.** To a major extent, the reasons behind the fully-connected Transformer baselines being able to excel in the proposed LRGB datasets can be linked to one or all of the characterizing factors that were discussed in Section 2. For instance, the nature of task of `Peptides-*` datasets, along with their substantial graph statistics (*i.e.*, avg. nodes, avg. shortest paths and avg. diameter as reported in Table 2) can help explain how long range dependencies are a must to do well on such tasks. Similarly, for `PCQM-Contact`, even if the graph sizes are small, the task of predicting pairs of distant nodes makes it a suitable LRGB dataset as shown by, e.g., Test Hits@3 scores of SAN against the local MP-GNNs in Table 5.

**Challenges and future directions.** First, the use of positional encoding alone contributes to little or no gain in performance on the proposed datasets. See the scores of GatedGCN augmented with LapPE or RWSE in Tables 3, 4 and 5 to this end. We hope such results to influence further exploration of powerful approaches to incorporate global structural and positional encoding in LRI enabled models, where the proposed LRGB can be used to conveniently evaluate the novel approaches. Second, the scores against each performance metrics in Tables 3-5 exhibit the current limitations of Transformers for graph learning and suggest that there is still a large window to fulfil by the better design of Graph Transformers that can make use of irregular sparse structure information, as well as propagate long range interactions. Finally, it must be noted that as we proceed towards evaluating Graph Transformers on long range benchmarks, such as our proposed LRGB with up to 479.40 avg. nodes, 58.79 million total nodes and 332.09 million total edges in a dataset, trivial $O(N^2)$ Transformers may be computationally inefficient to scale. To this end, research is also imperative on efficient or linear Transformers for graphs.

## 5   Conclusion

In this paper, we present the Long Range Graph Benchmark (LRGB) consisting of 5 datasets for node, edge and graph-level prediction tasks. Through our study of multiple characterizing factors, we argue that the proposed datasets' size and tasks makes these ideal to evaluate and develop models enabled with long-range dependencies. This is empirically verified with extensive baseline experiments using both local and non-local GNN classes showing that Transformers significantly outperform message passing on the proposed datasets. The increasing interest in the development of Transformers for graph representation learning raised the need for the creation of a dedicated LRGB and we fulfil this gap through our work. We believe our proposed benchmark can be leveraged to prototype new ideas and provide an accurate ranking of a model's capturing of LRIs.

## Acknowledgments and Disclosure of Funding

This work was partially funded by IVADO (Institut de valorisation des données) grant PRF-2019-3583139727 and Canada CIFAR AI Chair [*G.W.*]. Ali Parviz is supported by NSF Award #2039863. This research is supported by Nanyang Technological University, under SUG Grant (020724-00001). The content provided here is solely the responsibility of the authors and does not necessarily represent the official views of the funding agencies.

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
