# OpenReview forum: "Long Range Graph Benchmark"
_NeurIPS.cc/2022/Track/Datasets_and_Benchmarks — NeurIPS 2022 Datasets and Benchmarks _

### Official Review · Reviewer_WRaE · 2022-07-05
**Interesting paper, but the proposed dataset seems not suitable to evaluate the GNN's capability of capturing long-range interaction in the graph. The settings of the baseline are not convincing and lead to the unconvincing value of the dataset.**

**Rating:** 5
**Confidence:** 4

**Strengths:**

The benchmark dataset contains diverse tasks. They are derived from the real-world dataset. Most of the dataset contains graphs with large nodes and large diameters. The paper is timely. The GNN field lacks such a benchmark to test long-range interaction. The description of datasets is complete and easy to understand.

**Weaknesses:**

There are two major problems in the paper.

## Problem 1: I wonder whether the five proposed datasets can be served as testing the GNNs' capability of capturing large-range interaction in graphs.

There are many numbers on the paper indicating the size of the graph is large, or the diameter of them is large. However, they are not guaranteed that the dataset is useful for evaluating the GNNs' capability of capturing long-range interaction, as discussed in Section 2.

"However, this condition of graph size alone may not be enough to determine a LRGB dataset as the problem radius r may be small for some tasks even if the graph size is large, which brings us to the following factors."

So, if the graph is large and the diameter of the graph is large, the nature of the task is of critical importance in judging whether they are useful or effective to judge whether they are capable of evaluating GNNs' capability of capturing long-range interaction. As illustrated in Section 2:

"A dataset where the learning task benefits from global PE can hence be a potential LRI benchmark. Similarly, if the learning task in a dataset is dependent on some form of distance information or is directly a function of distance, coupled with graph feature information, the dataset can be a strong candidate for LRGB since the distance information would require global structural information. Examples of this can be molecular datasets where the learning task is related to the prediction of 2D or 3D distance and structure properties."

I somehow agree with this statement. However, in terms of the experimental results on these five tasks, the performance between GatedGCN and GatedGCN + global PE are quite comparable. It is negative evidence indicating that the constructed dataset is not suitable to serve a potential LRI benchmark.

Checking the description of each dataset further confirms such a negative impression.

- For example, the PASCAL VOC-SP and COCO-SP are formulated as the node-classification dataset. For semantic segmentation, CNN is somehow a popular network, and it is very good to capture local information. Similarly, the GNN used in those tasks can leverage the local neighborhood to predict the label of each superpixel without considering the long-range interaction.
- In terms of the PCQM-Contact dataset, it is formulated as a link prediction task. When computing whether two distant atoms are linked, it will be possible to extract the neighborhood of those two atoms and compute the similarity between their embedding of GNNs. It is unclear whether the final label depends on the distance between them or just based on the neighborhood of those atoms or similar or specific combination properties of those atoms. If the latter is true, the task will be independent of the distance between them and lack the ability to test the capability of capturing long-range interaction.
- In terms of Peptides molecular graphs, they are formulated as a graph classification task. Similarly, the final embedding of GNNs will consider the embedding of all nodes in the graph. It can also depend on the bag of properties of atoms without considering long-range interaction between them.

The authors may claim that the baseline results may mitigate such concerns. But current baseline settings are also worrying (Problem 2). Besides using baseline results to demonstrate the effectiveness of datasets, I recommend adding some concrete examples to battle against the candidate tricks proposed above and manually sample some examples (e.g., 100) from the dataset to check whether the above tricks can work. Moreover, concrete examples can be illustrated in a counterfactual style. For example, in terms of node classification tasks in the paper, you can demonstrate that modifying the features in a region of the image will change the labels of the distant superpixel.


## Problem 2: The number of layers used in MP-GNN is quite large.

When the number of layers is large (e.g., 5 or 8), it is possible that the MP-GNN suffers from over smoothing issues, as mentioned in the paper. The typical setting of the number of layers is two. If the authors can include the baseline results with a number of layers of two, it will be more convincing. Otherwise, I do not think that the performance of baseline models reflects their real capability to solve the task. I mean that the current performance of MP-GNN may be significantly lower than MP-GNN with two layers. If such a situation occurs, it is not good to say that it reveals the gap between local GNNs and non-local GNNs or that the dataset can be served as testing the GNNs' capability of capturing long-range interaction.

## Minor issues:

1. The citation cannot be served as a noun. Please fix it throughout the paper.
Line 36: "… are used, [2] used a fully …" should be modified as "… are used, Alon et al. [2] used a fully …".


**Additional Feedback:**

See weakness.

**Clarity:**

The paper is easy to read and follow. The contents are described and organized well.

**Correctness:**

There are some concerns on the settings of baselines and tasks used in the benchmark.

**Documentation:**

The appendix includes many details on the dataset.

**Relation To Prior Work:**

It follows the previous paper, "Benchmarking graph neural networks," and gives a new angle on long-range interaction.

**Summary And Contributions:**

The current benchmark on graph neural networks lacks testing the capability of capturing long-range interaction in graphs. In this paper, they present five datasets to benchmark the GNN's ability to capture long-range interaction. They conduct experiments with baseline and graph transformers to demonstrate that the benchmark can reveal the difference in different GNNs ability to capture long-range interaction.

---

> ### Author Response · Authors · 2022-08-22
> **Response to Reviewer WRaE**
>
> We thank the reviewer for her/his detailed comments and feedbacks on our work. We address the remaining concerns using the following responses.
>
> -----
> >**Reviewer**: Problem 1: I wonder whether the five proposed datasets can be served as testing the GNNs' capability of capturing large-range interaction in graphs.
>
> **Authors**: It is true, as highlighted in the review, that the proposed datasets are not guaranteed to require LRIs and it would be difficult to present a proof-included presentation of real-world LRI datasets.
>
> First, we would like to re-emphasize the discussion of characterizing factors in Section 2 and how the proposed datasets satisfy one or more of the factors that led us to propose them as LRGB dataset.
>
> Second, we kindly point the reviewer to the additional set of experiments (see also “Summary reply to all reviewers” above) –
> (i) Experiments with shallow MP-GNNs demonstrate that the proposed datasets are more suitable for evaluating GNNs with deeper   architectures, increased receptive fields, as well as long-range modeling because of stark difference in shallow (L=2) and deeper (L=5 or 8) models.
> (ii) Inspection of attention scores of Transformers shows that Transformer exhibits attention patterns beyond local neighborhoods in general compared to a similar study with MNIST superpixel dataset which has the contrasting observation.
>
> The two sets of experiments are included in Appendices D and E, respectively, in the revised manuscript.
>
> Third, as suggested by the reviewer for the task in Peptides-* datasets, we find that the ‘bag of atoms’ models are significantly underperforming (see table below for MLP compared with MP-GNNs) for which we trained and tested a simple MLP baseline (with the same L as other MP-GNNs) that operates without any information of graph structure. For questions on other datasets, we refer to the study and analysis in Appendix E.
>
>
> |  Models | Params | Peptides-func Test AP $\uparrow$ | Peptides-struct Test MAE $\downarrow$|
> |---|--:|:--:|:--:|
> | MLP |  506k | 0.4060±0.0021 | 0.4351±0.0008 |
> | GCN | 508k | 0.5930±0.0023 | 0.3496±0.0013 |
> | GCNII | 505k | 0.5543±0.0078 | 0.3471±0.0010 |
> | GINE | 476k | 0.5498±0.0079 | 0.3547±0.0045 |
> | GatedGCN | 509k | 0.5864±0.0077 | 0.3420±0.0013 |
> | GatedGCN+RWSE | 506k | 0.6069±0.0035 | 0.3357±0.0006 |
>
> Fourth, for the question on the contribution of global PE, see “Common concern 1” in our summary reply where we establish that global PE indeed provides critical structural information beyond closer neighbors when we experiment with shallow MP-GNNs. We hope it helps clarify that our statement of “A dataset where the learning task benefits from global PE can hence be a potential LRI benchmark” is consistent accordingly and we believe that we have no negative evidence in this regard.
>
> -----
> >**Reviewer**: Problem 2: The number of layers used in MP-GNN is quite large.
>
> **Authors**: Please see “Common concern 1” in our summary reply.
>
> -----
> We hope that the aforementioned revision and added deeper investigations support the proposed datasets in being suitable for testing long-range reasoning architectures, and are a substantial contribution to the community where such new models are still evaluated on traditional general-purpose benchmarks.
>
> -----
> >**Reviewer**: Minor issues: 1. The citation cannot be served as a noun. Please fix it throughout the paper. Line 36: "… are used, [2] used a fully …" should be modified as "… are used, Alon et al. [2] used a fully …".
>
> **Authors**: Thank you for the feedback. We fixed the formatting throughout the manuscript.

---

> > ### Comment · Reviewer_WRaE · 2022-08-24
> > **Further clarification is needed.**
> >
> > I appreciate the authors response to provide more evidence. But I still have some concerns. If the following concerns can be addressed, I will raise my score.
> >
> > ## Problem 1: I wonder whether the five proposed datasets can be served as testing the GNNs' capability of capturing large-range interaction in graphs.
> >
> > I am not expecting that you need to prove each examples in the dataset require LRIs. What I expect is that one concrete example can be illustrated in a counterfactual style. For example, in terms of node classification tasks in the paper, you can demonstrate that modifying the features in a region of the image will change the labels of the distant superpixel. You can use some GNN claimed to capture LRIs to monitor whether modifying the features in a region of the image will change the labels of the distant superpixel. One case for each task is enough to demonstrate that the task itself contains some cases require LRIs. Reasonable justifications for changing the labels should also be provided.
> >
> > ## Problem 2: The number of layers used in MP-GNN is quite large.
> >
> > Controlling the budget is good. But why you make sure all those models have the same parameters via changing the dimensions of hidden weights? Instead, I expect that the dimension should be the same as previous experiments (5 or 8 layers), regardless of budget control, rather than increasing the dimensions to ensure the same number of parameters. For simple models, more parameters are not guaranteed that the performance will be better as the probability of overfitting is also increasing.

---

> > > ### Author Response · Authors · 2022-08-27
> > > **Further response (1/2) to Reviewer WRaE**
> > >
> > > >**Reviewer**: Problem 2: The number of layers used in MP-GNN is quite large.
> > > Controlling the budget is good. But why you make sure all those models have the same parameters via changing the dimensions of hidden weights? Instead, I expect that the dimension should be the same as previous experiments (5 or 8 layers), regardless of budget control, rather than increasing the dimensions to ensure the same number of parameters. For simple models, more parameters are not guaranteed that the performance will be better as the probability of overfitting is also increasing.
> > >
> > > **Authors**:
> > > Re. “why you make sure all those models have the same parameters”: We would like to point to Section 4.1 (lines 331-333) and clarify that this choice is made in all the experiments to ensure fair comparison of all baselines. Although this choice can be debated, we believe that it is a just technique that can be used to compare a zoo of models with similar learning capacity in terms of model parameters. Since our paper focuses on ‘baseline benchmarking’ and not ‘proposing any model to achieve SOTA’, we feel that the question towards this ‘model comparison in the same budget’ to be insignificant and should not affect the central claims and contributions brought by the paper. We nevertheless acknowledge your comment and concerns on this point, and therefore we additionally conducted the L=2 experiments again, now with the same hidden dimensions as that of L=5,8. The results are included below in this response, compared to that of L=2 with 500k parameter budget.
> > >
> > >
> > > |  Models (L=2) | Params | Peptides-func Test AP $\uparrow$ | Peptides-struct Test MAE $\downarrow$|
> > > |---|--:|:--:|:--:|
> > > | GCN | 236k | 0.4536±0.0063 | 0.3962±0.0012 |
> > > | GCN | 509k | 0.4566±0.0059 | 0.3950±0.0017 |
> > > | GCNII | 234k | 0.4816±0.0041 | 0.3961±0.0025 |
> > > | GCNII | 507k | 0.4894±0.0039 | 0.3929±0.0020 |
> > > | GINE | 214k | 0.4965±0.0032 | 0.3883±0.0005 |
> > > | GINE | 501k | 0.5003±0.0042 | 0.3879±0.0011 |
> > > | GatedGCN | 219k | 0.5038±0.0018 | 0.3909±0.0009 |
> > > | GatedGCN | 508k | 0.5073±0.0036 | 0.3905±0.0006 |
> > > | GatedGCN+RWSE | 216k | 0.5657±0.0074 | 0.3611±0.0002 |
> > > | GatedGCN+RWSE | 505k | 0.5812±0.0053 | 0.3599±0.0007 |
> > >
> > > From the above results, it can be observed that *in all the comparisons* the models with L=2 and 500k params are *marginally better* than those with L=2 and fixed width. This is not surprising as the two model configurations are not that much different, with the only change being the hidden representation size. We hope this additional set of experiments and comparison clarifies your concern on this comment. Please note that we limit these additional experiments to 2 out of 5 proposed datasets (Peptides-*), keeping in consideration the time remaining for the discussion period, and expect the observation to be consistent on all other datasets.
> > >
> > > Next, we believe we have already demonstrated the effect of having a smaller number of layers in MP-GNNs for our baseline experiments by the added experiments in the revision. We reiterate our observation of L=2 MP-GNNs performing poorly when having fewer number of layers, which demonstrates the datasets proposed in the paper are quite different from traditional benchmarks and a small receptive field for the proposed tasks is a considerable limitation. With this revision already done, we feel that we have sufficiently addressed your concern on the “effect of oversmoothing when having L=5,8 for MP-GNNs”. Please refer to lines 352-359 in Section 4.2 in the revised manuscript.

---

> > > > ### Author Response · Authors · 2022-08-27
> > > > **Further response (2/2) to Reviewer WRaE**
> > > >
> > > > >**Reviewer**: Problem 1: I wonder whether the five proposed datasets can be served as testing the GNNs' capability of capturing large-range interaction in graphs.
> > > > I am not expecting that you need to prove each examples in the dataset require LRIs. What I expect is that one concrete example can be illustrated in a counterfactual style. For example, in terms of node classification tasks in the paper, you can demonstrate that modifying the features in a region of the image will change the labels of the distant superpixel. You can use some GNN claimed to capture LRIs to monitor whether modifying the features in a region of the image will change the labels of the distant superpixel. One case for each task is enough to demonstrate that the task itself contains some cases require LRIs. Reasonable justifications for changing the labels should also be provided.
> > > >
> > > >
> > > > **Authors**: Thank you for the suggestion of demonstrating a counterfactual example to show that the datasets indeed require LRI enabled models to do well. We understand this point to be one of several other suggestions/remarks presented in your original review above under the heading ‘Problem 1’. We believe that we have already addressed most of your concerns of ‘Problem 1’ except this one, which we think may not be a robust enough way to show the proposed datasets’ LRI properties. Because one (or a handful) example for each task, as your suggestion refers, could be any random example graph, these could be examples that fall in either of positive or negative evidence categories. Therefore, we understand that even if we show a handful of positive example cases for this demonstration, it could appear as cherry picking and would not serve as a robust and fair demonstration. Although the suggestion is well intended and we are thankful for that, it would eventually resemble a small qualitative result, which may not elucidate the general characteristic of an entire dataset. Nevertheless, we would like to point again to the study presented in Section E, as recommended by Reviewer 9aLb, in which we perform a similar interpretation/demonstration of long range interaction using Transformer. Being LRI enabled, Transformer *generally* exhibits attention patterns beyond local neighborhoods, which is an answer to your question albeit using a slightly different approach.
> > > >
> > > > Having said that, we strongly feel that there are other reasons, properties, experiments, visualizations in the paper, including after the revision during this discussion which show the proposed datasets to be good candidates for long-range graph benchmarking. We would appreciate it if you could consider the merits of our work along all these lines and the addressal of other concerns except for this counterfactual demonstration.

---

### Official Review · Reviewer_mHcB · 2022-07-18
**Both evaluation metrics and baselines are insufficient.**

**Rating:** 4
**Confidence:** 4
**Correctness:** Yes
**Clarity:** Yes

**Strengths:**

Pros.

1. Diverse tasks and graph properties are included in the proposed benchmark.

2. Sufficient literature is provided.

3. Visualizations are nice like figure 3.


**Weaknesses:**

Cons.

1. It is necessary to also benchmark the cost like FLOPS and running time. Otherwise, it is unfair to conclude that Graph Transformer is better.

2. As shown in Table 2, why 10-50 avg. diameter is enough to indicate the property of the long-range relationship? Need to compare with common graphs.

3. Since the over-squashing issue is one of the key problems in this paper, it is necessary to reveal how long-range or how spare the connection will result in over-squashing.

4. The authors mention "we believe there is consensus in the community towards the development of specific benchmark that can assist LRI enabled-GNNs". Need more citations to support it.

5. More classic GNN backbones like SGC and GCNII are required.

6. Need to provide more analyses with deep GCNs since the depth can help capture the long-range relationship.


**Additional Feedback:**

Refer to the weakness section.

**Documentation:**

Yes

**Ethics:**

I do not see any ethical concerns.

**Relation To Prior Work:**

Yes

**Summary And Contributions:**

Summary.

The paper proposes the long-range graph benchmark (LRGB) with 5 graph learning datasets. The core motivation is that: MP-GNNs mainly reply on 1-hop message passing which may limit its expressiveness. CNN and Graph Transformer networks are evaluated on these proposed datasets.

---

> ### Author Response · Authors · 2022-08-22
> **Response to Reviewer mHcB**
>
> We thank the reviewer for her/his comments and suggestions on our paper. We clarify and address the feedback as follows.
>
> -----
> >**Reviewer**: Weaknesses: 1. It is necessary to also benchmark the cost like FLOPS and running time. Otherwise, it is unfair to conclude that Graph Transformer is better.
>
> **Authors**: We now include the running times in Appendix C, Table C.2, please see “Common concern 3” in our summary reply.
>
> -----
> >**Reviewer**: Weaknesses: 2. As shown in Table 2, why 10-50 avg. diameter is enough to indicate the property of the long-range relationship? Need to compare with common graphs.
>
> **Authors**: We would like to clarify that we do not claim that the mentioned avg. diameter is enough but rather it is “one of” the characterizing factors as discussed in Section 2. In fact, we precisely articulate this statement in lines 114-116. Re., comparing with common graphs, we have mentioned the respective ‘existing relevant graph dataset’ comparison in terms of avg. diameter in Section 3.1 (w/ MNIST and CIFAR10 datasets), and in Section 3.3 (w/ ENZYMES and PROTEINS datasets). As for other existing datasets, in Section 1, we do not directly show the avg. diameter but highlight the small sizes of ZINC, ogbg-molhiv and ogbg-molpcba which effectively means their avg. diameter are smaller.
>
> -----
> >**Reviewer**: Weaknesses: 3. Since the over-squashing issue is one of the key problems in this paper, it is necessary to reveal how long-range or how spare the connection will result in over-squashing.
>
> **Authors**: We refer to the second and third paragraphs in Section 1 where we discuss the problem of over-squashing and how direct (full) connections of each node to all nodes in a graph is one way to overcome the problem, among others. On the other hand, sparse connections (by which we understand the local 1-hop connections as employed in MP-GNNs) will lead to the over-squashing of long-range information.
>
> -----
> >**Reviewer**: Weaknesses: 4. The authors mention "we believe there is consensus in the community towards the development of specific benchmark that can assist LRI enabled-GNNs". Need more citations to support it.
>
> **Authors**: We provide another reference from Alon et al. 2021 in Section 1, page 3 apart from the existing citations where synthetic, toy or semi-real datasets were used to demonstrate the need of LRI enabled GNNs. Besides, in the two paragraphs in Section 1, page 3 under the heading “Existing attempts towards LRI benchmark” we provide numerous references to methods which propose architectures that are intended to do well on long-range/non-local tasks but their models’ evaluations do not have benchmarks available for the same. For all these reasons, we believe that there is consensus in the community towards the development of a specific long range graph benchmark which we try to address through our work.
>
> -----
> >**Reviewer**: Weaknesses: 5. More classic GNN backbones like SGC and GCNII are required.
>
> **Authors**: We added GCNII experiments in the current revision, please see “Common concern 2” in our summary reply.
>
> -----
> >**Reviewer**: Weaknesses: 6. Need to provide more analyses with deep GCNs since the depth can help capture the long-range relationship.
>
> **Authors**: In principle we agree that deeper MP-GNNs could achieve better performance on several of the proposed datasets, as long as the potential over-smoothing and over-squashing do not become detrimental. We used 5 or 8 layer MP-GNNs with residual connections to alleviate the above issues, and additionally we conducted experiments with 2-layer models as well (please see “Common concern 1” in our summary reply), to show a difference between shallow and deep(er) architectures. Please note that even deeper models would need to further sacrifice hidden node representation size in order to fit within the 500k parameter budget, which also could lead to lower performance.

---

> > ### Author Response · Authors · 2022-08-27
> > **Follow up with Reviewer mHcB**
> >
> > Dear Reviewer mHcB,
> >
> > We thank you once again for your time in reviewing our work.
> >
> > As the discussion period is about to end, we kindly ask your response and feedback on our answers above in reply to your initial review. We hope we have addressed most of your concerns as we provide a revised manuscript incorporated with several suggestions provided in your initial review such as GCNII baseline, running times of models, and clarifications on avg. diameter, oversquashing, depth, etc.
> >
> > We would appreciate your feedback on our revision and whether it has addressed your concerns. If so, we would appreciate it if you could consider raising the score.
> >
> > With best regards!

---

### Official Review · Reviewer_9Uqd · 2022-07-21
**Review of Long Range Graph Benchmark**

**Rating:** 7
**Confidence:** 5
**Clarity:** The paper is well-written, and basica…

**Strengths:**

Long range dependency is an important issue of graph learning. Existing real-world datasets on graph learning generally does not focus on these point. In this sense, this paper acts as a complement to the existing datasets and can pose the challenges to the current popular graph learning models.

**Weaknesses:**

This paper restricts the models that can learn from the long-range dependencies to the transformer ones. Actually there are many other graph learning models which can capture the long-range dependencies and are not discussed or evaluated in this paper. The authors are suggested to discuss these long-range graph learning models in this paper, such as:
Eignn: Efficient infinite-depth graph neural networks

**Additional Feedback:**

Please refer to the suggestions given above.

**Correctness:**

Although the authors claim that "our aim is to propose real-world datasets that require LRI ... would depend on long range signal propagation." In the detailed descriptions of the task, construction, etc. of different datasets. Why the specific datasets need the long-range dependencies are not explained or highlighted clearly. The authors are suggested to emphasize the importance of long-range dependencies on different datasets in more detail.

**Documentation:**

There is sufficient detail on data collection and organization, availability and maintenance, and ethical and responsible use. The URL for reviewer access to the dataset is available.

**Ethics:**

The authors claim that they have read the ethics review guidelines and ensured that their paper conforms to them.

**Relation To Prior Work:**

The authors are suggested to discuss these long-range graph learning models in this paper, such as:
Eignn: Efficient infinite-depth graph neural networks

**Summary And Contributions:**

In this work, the authors focus on these shortcomings of existing popular graph learning benchmark datasets and propose characterizing factors in a dataset that can be studied for the exploration of new GNN and Graph Transformer architectures that possess long-range interaction (LRI) capabilities. The authors' aim is to propose real-world datasets that require LRI, and the factors they consider for a graph dataset characterization could be understood as implications that suggest that the learning task(s) in the graphs would depend on long-range signal propagation. Consequently, the authors introduce 5 benchmarking datasets – PascalVOC-SP, COCO-SP, PCQM-Contact, Peptides-func and Peptides-struct from the domains of Computer Vision and Chemistry which we incorporate in LRGB. The learning tasks that the authors propose in these datasets depend on some degree of long-range signal handling given the nature of task, contribution of global graph structure to the task, and the sizes of graphs in these datasets. Fittingly, in the authors' baseline experiments, these datasets show that the fully-connected models which enable LRI propagation perform better than local message passing-based GNNs.

---

> ### Author Response · Authors · 2022-08-22
> **Response to Reviewer 9Uqd**
>
> We thank the reviewer for highlighting the contribution of our work and providing useful feedback which we address with the following answers.
>
> -----
> >**Reviewer**: Weaknesses: This paper restricts the models that can learn from the long-range dependencies to the transformer ones. Actually there are many other graph learning models which can capture the long-range dependencies and are not discussed or evaluated in this paper. The authors are suggested to discuss these long-range graph learning models in this paper, such as: Eignn: Efficient infinite-depth graph neural networks
>
> **Authors**: Please see “Common concern 2” in our summary reply.
>
> -----
> >**Reviewer**: Correctness: Although the authors claim that "our aim is to propose real-world datasets that require LRI ... would depend on long range signal propagation." In the detailed descriptions of the task, construction, etc. of different datasets. Why the specific datasets need the long-range dependencies are not explained or highlighted clearly. The authors are suggested to emphasize the importance of long-range dependencies on different datasets in more detail.
>
> **Authors**: We would like to clarify that it would be difficult to present a proof-like depiction of whether the real datasets we have proposed require LRIs. We present a discussion to this aspect in the phrases you quote (lines 60-65) that led to the characterization in Section 2. However, where possible, we have tried to articulate the factors that would suggest either ‘satisfying characteristics’ or ‘necessity of long range signal propagation for the proposed tasks’. For instance, lines 159-162 for PascalVOC-SP which follows for COCO-SP as well, and Task descriptions in PCQM and Peptides-struct. In case of Peptides-func, there are multi-labels for a graph, and it would be difficult to pinpoint which specific task requires long-range dependencies, yet the characterizing factors and empirical results show that it is also a suitable LRGB dataset.
>
> Beyond the above argumentation, to provide further empirical evidence for suitability of the proposed LRGB datasets, we now include performance of MP-GNNs with just 2 layers in Appendix D, please see “Common concern 2” in our summary reply, and also inspection of attention distributions in Transformer with LapPE (Appendix E).

---

### Official Review · Reviewer_PA7X · 2022-07-25
**Interesting benchmark to graph learning**

**Rating:** 6
**Confidence:** 4
**Correctness:** Good.
**Clarity:** Good.

**Strengths:**

1. Nice discussion to motivate the benchmark
2. Preparation of diverse tasks/datasets on LRI evaluation
3. Experiment with multiple methods

**Weaknesses:**

I think the result analysis part could be improved. For example, in Table 3, we can see GatedGCN performs better than  Transformer+LapPE but worse than SAN on PascalVOC-SP. Then how is the SAN's modeling help LRI could be discussed and further experimented. Same for other datasets. The transformer methods are not significantly better and could be explored.

Lacking such analysis would limit people's interest in using the benchmark and further contribute to the community.

**Additional Feedback:**

None.

**Documentation:**

Good.

**Ethics:**

Good.

**Relation To Prior Work:**

Good.

**Summary And Contributions:**

Long Range Interaction is becoming an interesting topic in graph learning community thus a good benchmark is needed. This paper proposes a benchmark of 5 datasets whose diameters are larger than usual graph datasets and the authors experiment different methods based upon. They conclude that the fully-connected models which enable LRI propagation perform considerably better than local message passing based GNNs.

---

> ### Author Response · Authors · 2022-08-22
> **Response to Reviewer PA7X**
>
> We thank the reviewer for highlighting the strengths of our work on the lines of motivation, diversity of datasets in the proposed benchmark and the experiments. We also thank the reviewer for the valuable feedback raised which we address with the following.
>
> -----
> >**Reviewer**: Weaknesses: I think the result analysis part could be improved. For example, in Table 3, we can see GatedGCN performs better than Transformer+LapPE but worse than SAN on PascalVOC-SP. Then how is the SAN's modeling help LRI could be discussed and further experimented. Same for other datasets. The transformer methods are not significantly better and could be explored.
>
> **Authors**: Re. Table 3, the performance of Transformer+LapPE compared to GatedGCN is indeed a negative result which we include in the revision of Section 4.2 (line 362). One probable reason behind this is the strength of GatedGCN, an attention-like MP-GNN, which suffers from over-squashing to a lesser extent compared to GCN and GIN. We have already included this analysis in the lines 350-351 in the manuscript. Regarding other datasets’ results and the ability of SAN in performing better, we argue how the consideration of non-local connections in graphs in non-trivial manner in SAN leads to better capturing of non-local information, see lines 365-367. Finally, we expose the limitations of the existing Transformer methods through our experimental analysis in the lines 385-389 where we pose that a lot could be explored as future research to fulfill the gap of performance.

---

### Official Review · Reviewer_ckvd · 2022-07-26
**Complete long range graph datasets and empirical benchmark results.**

**Rating:** 7
**Confidence:** 4
**Correctness:** Yes. The claims, dataset construction…
**Clarity:** Yes.

**Strengths:**

(1) The motivation of this work is grounded and well justified. How to capture long range interactions over graphs is an important and under-explored problem. This work identifies a good problem and provides a convincing testbed for future work.

(2) The proposed datasets are well processed and presented. They cover different sizes and domains, which are comprehensive for evaluating future work.

(3) The empirical benchmark results are complete and statistically valid.

(4) The presentation flow of this paper is clear and easy to follow.


**Weaknesses:**

(1) This work demonstrates the effectiveness of transformer-like models over local GNNs. However, as discussed in this paper, the computational complexities of the transformer-like models are much higher. As a benchmark work, it would be more rigorous to compare the empirical running time of these two types of models.

**Additional Feedback:**

(1) There are a bunch of works learning from non-homophilous graphs, such as Geom-GCN [1] and non-local GNN [2]. Although it is a different problem with the one considered this work, it is quite related. It is also desired to capture the long-range information, since the graph is non-homophilous. I think the difference with this problem should be clarified. Also, I am wondering how these methods perform on the datasets proposed in this work.



[1] Pei, Hongbin, et al. "Geom-gcn: Geometric graph convolutional networks." arXiv preprint arXiv:2002.05287(2020).

[2] Liu, Meng, Zhengyang Wang, and Shuiwang Ji. "Non-local graph neural networks." IEEE Transactions on Pattern Analysis and Machine Intelligence (2021).


**Documentation:**

Yes, it is well formulated on the GitHub page.

**Ethics:**

None.

**Relation To Prior Work:**

Partially. See the additional feedback part.

**Summary And Contributions:**

This work develops five datasets for studying long range interactions on graphs. The proposed datasets are well clarified and processed. In addition, experiments of local and non-local models have been conducted over these datasets to justify the motivation of this work.

---

> ### Author Response · Authors · 2022-08-22
> **Response to Reviewer ckvd**
>
> We thank the reviewer for careful reading and comments on our work and showing the strengths of the benchmark in terms of motivation, diversity of the datasets, empirical results and the paper presentation. We address the raised questions with our following answers.
>
> -----
> >**Reviewer**: Weaknesses: (1) This work demonstrates the effectiveness of transformer-like models over local GNNs. However, as discussed in this paper, the computational complexities of the transformer-like models are much higher. As a benchmark work, it would be more rigorous to compare the empirical running time of these two types of models.
>
> **Authors**: Please see “Common concern 3” in our summary reply.
>
> -----
> >**Reviewer**: Additional Feedback: (1) There are a bunch of works learning from non-homophilous graphs, such as Geom-GCN [1] and non-local GNN [2]. Although it is a different problem with the one considered this work, it is quite related. It is also desired to capture the long-range information, since the graph is non-homophilous. I think the difference with this problem should be clarified. Also, I am wondering how these methods perform on the datasets proposed in this work.
> [1] Pei, Hongbin, et al. "Geom-gcn: Geometric graph convolutional networks." arXiv preprint arXiv:2002.05287(2020).
> [2] Liu, Meng, Zhengyang Wang, and Shuiwang Ji. "Non-local graph neural networks." IEEE Transactions on Pattern Analysis and Machine Intelligence (2021).
>
> **Authors**: Please see “Common concern 2” in our summary reply.

---

> > ### Comment · Reviewer_ckvd · 2022-08-24
> > **Thanks**
> >
> > Thanks for the response. I would like to keep my score as accept.

---

### Official Review · Reviewer_9aLb · 2022-07-27

**Rating:** 7
**Confidence:** 4
**Correctness:** The dataset construct in reasonable.
**Clarity:** The paper is well written and clear e…

**Strengths:**

1. Enough samples and variety of different types of graphs, which will attract researchers from different research fields.
2. The evaluation is sufficient to show that the learning on the proposed graph needs long range information from neighbors.
3. The label and graph is clear and easy to understand.

**Weaknesses:**

1. It will be better to show the performance of models with fewer layers and broader widths, which could further support the claim that learning requires long-range information.
2. The learning rate is important for some graph learning tasks. I wonder if the authors searched that. The conclusion could be different.
3. Since the transformer is used in the evaluation. Is it possible to visualize the attention score, i.e., where does the network pay attention in classification? It would be better if the attention scores also supported the claim.
4. Not all message passing networks use 1-hop neighbors, e.g., Digraph Inception Convolutional Networks, NeurIPS 2020.

**Additional Feedback:**

I appreciate the authors' effort in data construction and comparison, and I am inclined to raise the score if the authors could address my questions.

**Documentation:**

It would be better if more details could be available in Github. For now most information is contained in the paper.

**Ethics:**

There is no ethical concern.

**Relation To Prior Work:**

Yes. The discussion is included about what's the conventional setting and the long range settings.

**Summary And Contributions:**

This paper proposed a set of datasets for long range graph learning. The datasets include sufficient samples and detailed construction steps. The evaluation supports the claim that the tasks requires long range information.

---

> ### Author Response · Authors · 2022-08-22
> **Response to Reviewer 9aLb**
>
> We thank the reviewer for the comments regarding our work, particularly highlighting the strengths of the proposed datasets, evaluation and usage. We address the raised concerns with the following answers.
>
> > **Reviewer**: Weaknesses 1. It will be better to show the performance of models with fewer layers and broader widths, which could further support the claim that learning requires long-range information.
>
> **Authors**: Please see “Common concern 1” in our summary reply.
>
> > **Reviewer**: Weaknesses 2. The learning rate is important for some graph learning tasks. I wonder if the authors searched that. The conclusion could be different.
>
> **Authors**: We did not perform thorough hyperparameter search (including the learning rate) for the MP-GNNs and Transformers presented as baseline models on the proposed datasets. Instead, we follow a fair strategy to compare these baselines under a budget of 500k model parameters with the details presented in Appendix C.1. We would like to highlight that we do not use a constant learning rate throughout the training procedure, but rather a learning rate decay mechanism that decreases the learning rate when the validation score plateaus, as mentioned in Appendix C.1. Nevertheless, our attempt is to not restrict the capabilities of any baseline model presented in the paper with an ‘adverse’ initial learning rate and the above strategies allow for adequate model convergence during training within the stipulated time limit.
>
> >**Reviewer**: Weaknesses 3. Since the transformer is used in the evaluation. Is it possible to visualize the attention score, i.e., where does the network pay attention in classification? It would be better if the attention scores also supported the claim.
>
> **Authors**: We conducted the requested experiments and present them in Appendix E.  In particular, we investigate how strongly a Transformer attends to nodes that are at various $k$ distances away from a node $v$ during updating of its representation $h_v^\ell$ at layer $\ell \in \{0, \dots, L-1\}$. The goal is to probe whether a model capable of global attention, such as the Transformer with LapPE, in fact attends to nodes farther than the local neighborhood of $v$, while performing better or comparable to local MP-GNN models. Overall the attention distributions vary across datasets and layers, but generally confirm that Transformer exhibits attention patterns beyond local neighborhoods. Please see Appendix E for further detail.
>
> >**Reviewer**: Weaknesses 4. Not all message passing networks use 1-hop neighbors, e.g., Digraph Inception Convolutional Networks, NeurIPS 2020.
>
> **Authors**:
> We clarify this detail by revising the phrase ‘.. exchange information .. ’ to ‘.. *generally* exchange information .. ’ in line 2 of the manuscript. Please also see “Common concern 2” in our summary reply.
>
> >**Reviewer**: Documentation: It would be better if more details could be available in Github. For now most information is contained in the paper.
>
> **Authors**: Thank you for the recommendation. We added more details on the dataset in the documentation of the GitHub README.
>
> >**Reviewer**: Additional Feedback: I appreciate the authors' effort in data construction and comparison, and I am inclined to raise the score if the authors could address my questions.
>
> **Authors**: We thank you for your suggestions and appreciation of our work. We hope we have clarified your concerns with the additional experiments, appropriate writing revision and due clarification with our answers above. If so, please consider raising the score, or let us know of any outstanding concerns!

---

> > ### Comment · Reviewer_9aLb · 2022-08-24
> > **Thanks for the response!**
> >
> > The authors addressed my concerns, so I increased the score.

---

### Author Response · Authors · 2022-08-22
**Summary reply to all reviewers**

We thank all reviewers for their time and thorough reviews! We carefully considered all the points raised and we answer the most common questions in this summary reply. Please note that we have uploaded a revised version of the paper (using a format with line number annotations for ease of reference to a particular text). We look forward to further discussion, and if we answered your concerns, please consider raising your score if appropriate.

------
>**Common concern 1**: The benchmarked baseline MP-GNNs are too deep (5 or 8 layers), this can lead to over-squashing or over-smoothing, giving unfair advantage to Transformer-based models. Further, a gap between shallow and deep MP-GNNs could empirically indicate that limited local neighborhood aggregation is not sufficient and long(er)-range interactions are important.


**Authors**: We conducted suggested experiments with shallower but wider model architectures. We present the performance of MP-GNN models with 2 layers and ~500k parameters in Appendix D. Generally, we observe majorly decreased performance of 2-layer MP-GNNs as compared to their deeper versions, while their relative ordering by their performance remains largely the same. This finding confirms that access to only a narrow receptive field is severely limiting. Additionally, we observe much increased positive impact of augmenting 2-layer GatedGCN with positional or structural encodings. GatedGCN with LapPE or RWSE outperforms standard GatedGCN (and any other tested MP-GNN) by a large margin particularly in PCQM-Contact, Peptides-func, and Peptides-Struct. In the case of the deeper MP-GNN configurations this effect is not observed, suggesting that the positional or structural encodings provide additional information beyond the 2-hop neighborhood that a deeper GatedGCN appears to be able to substitute.

------

>**Common concern 2**: There are several more models, other than the graph Transformer-based, that perform non-local aggregation or are capable of capturing long(er)-range interactions in heterophilic graphs; e.g., Geom-GCN, non-local GNN, EIGNN; or could serve as interesting baselines, e.g., SGC or GCNII. How would these methods perform on the proposed LRGB datasets?



**Authors**: We thank the reviewers for the references to these models. Upon a closer inspection we conclude that most of these methods are primarily intended for the transductive setting, i.e. learning in one large graph. While these methods could be adjusted to inductive tasks, the published codebases we found on GitHub do not support it out-of-the-box or would require non-trivial reimplementation in our PyG-based framework. Ultimately, the purpose of our work is to introduce new benchmarking datasets and showcase their applicability for the intended use. Providing a thorough benchmarking of many applicable models/approaches and their inspection is an appealing future work worthy of a separate paper, but outside of the current scope. Nevertheless, we added another commonly used MP-GNN baseline, the GCNII. All result tables have been extended and are part of the uploaded paper revision.

------

>**Common concern 3**: Empirical run-times of MP-GNNs and Transformer-based models.

**Authors**: For all benchmarked methods, we measured average epoch wall-clock times on a system with a single NVidia A100 GPU and assignment of 4 cores of AMD Milan 7413 CPU. Please see these results in Appendix C, Table C.2.

------

**Summary of the main changes in the revision**:
- Depositited the LRGB dataset in a tracked repository at Zenodo: https://doi.org/10.5281/zenodo.6975830.
- Added references to related prior work in the Introduction, as suggested by the reviewers.
- Added GCNII model in all results tables.
- Added Appendix D: Additional Experiments with L=2 MP-GNN models. Additionally, expanded Section 4.2 by a paragraph summarizing the shallow MP-GNNs results.
- Added Appendix E: Inspection of Transformer attention distribution.
- Added summary table of run times in Appendix C, Table C.2.
- Updated the GitHub repo README with datasets’ overview.
------

---

### Public Comment · ~Heng_Chang2 · 2022-11-30
**Missing discussion with a highly related work and some questions**

Hi, this is really an interesting work that benchmarks the long-range interaction capabilities of GNNs and Graph Transformers. The findings are informative with plentiful insights and empirical support.

We found that the authors seem to miss the discussion with an early and highly related work [1], where we study the ability of GNNs to capture long-range dependencies in underlying graphs with implicit models and propose a synthetic chains data set for evaluation accordingly. Some later works[2-4] also follow our setting for designing more powerful GNNs to improve this ability.

Maybe I missed something, but it seems like the long-range interaction and long-range dependency denote a very similar challenge in graph learning. Although it doesn't invalidate the main conclusions and contributions of the paper, could the authors offer some discussions on the two tasks?

[1] Gu et al., Implicit Graph Neural Networks, NeurIPS 2020.

[2] Yang et al., Graph neural networks inspired by classical iterative algorithms, ICML 2021

[3] Kim et el., Transformers Generalize DeepSets and Can be Extended to Graphs and Hypergraphs, NeurIPS 2021

[4] Liu et al., MGNNI: Multiscale Graph Neural Networks with Implicit Layers, NeurIPS 2022

---

### Meta-Review · Area_Chair_vsjr · 2022-09-11

**Recommendation:** Accept
**Confidence:** 4

**Metareview:**

In this paper, the authors present a benchmark for long-range graph learning. All reviewers found this to be well-motivated and appreciate that this work provides a diverse suite of tasks and covers a range of graph properties. During the discussion period, the authors further clarified several reviewers' comments.

The feedback from reviewers is mostly positive. There are several opportunities for improvement/clarification, especially for reviewers @WRaE and @mHcB -- it would be great if the authors can take them into consideration.

Overall, I would recommend this paper to be Accepted.

---

### Decision · Program_Chairs · 2022-09-16

Accept